

# The impact of IL-6 and IL-28B gene polymorphisms on treatment outcome of chronic hepatitis C infection among intravenous drug users in Croatia

Zoran Bogdanović[1],*, Ivana Marinović-Terzić[2],*, Sendi Kuret[3],*, Ana Jerončić[4],*, Nikola Bradarić[5], Gea Forempoher[3], Ozren Polašek[6], Šimun Anđelinović[3] and Janoš Terzić[2]

[1] Department of Internal Medicine, Division of Gastroenterology, Clinical Hospital Split, Split, Croatia
[2] Department of Immunology, Univeristy of Split, School of Medicine, Split, Croatia
[3] Department of Pathology, Clinical Hospital Split, Split, Croatia
[4] Department of Research in Biomedicine and Health, Univeristy of Split, School of Medicine, Split, Croatia
[5] Department of Infectious Diseases, Clinical Hospital Split, Split, Croatia
[6] Department of Public Health, Univeristy of Split, School of Medicine, Split, Croatia
* These authors contributed equally to this work.

Corresponding authors
Zoran Bogdanović,
suzbog@gmail.com
Janoš Terzić, janos.terzic@mefst.hr

## ABSTRACT

**Background:** Several genes and their single nucleotide polymorphisms (SNPs) are associated with either spontaneous resolution of hepatitis C infection or better treatment-induced viral clearance. We tested a cohort of intravenous drug users (IVDU) diagnosed with chronic hepatitis C virus (HCV) for treatment response and its association with the SNPs in the interleukin-6 (rs1800795-IL6) and the interleukin-28B (rs12979860-IL28B) genes.

**Methods:** The study included 110 Croatian IVDU positive for anti-HCV antibody. Genotyping was performed by polymerase chain reaction (PCR) based approach. Patients were treated by standard pegylated-interferon/ribavirin and followed throughout a period of four years, during which sustained virological response (SVR) was determined. All data were analysed with statistical package SPSS 19.0 (IBM Corp, Armonk, NY, USA) and PLINK v1.07 software.

**Results:** Patients showed a significantly better response to treatment according to the number of copies of the C allele carried at rs1800795-IL6 ($P = 0.034$). All but one of the patients with CC genotype achieved SVR (93%), whereas the response rate of patients with GG genotype was 64%. The association of rs1800795-IL6 with SVR status remained significant after further adjustment for patients' age, fibrosis staging, and viral genotype (OR 2.15, 95% CI 1.16–4.68, $P = 0.019$). Distributions of allele frequencies at the locus rs12979860-IL28B among the study cohort and the underlying general population were suggestive of a protective effect of CC genotype in acquiring chronic hepatitis C in the Croatian IVDU population.

**Discussion:** The rs1800795-IL6 polymorphism is associated with positive response to treatment in IVDU patients positive for HCV infection. A protective role of rs12979860-IL28B CC genotype in acquiring chronic hepatitis C is suggested for Croatian IVDU population.

## INTRODUCTION

It is estimated that chronic hepatitis C virus (HCV) infection affects nearly 170 million individuals worldwide (*Averhoff, Glass & Holtzman, 2012*; *World Health Organization, 2014*). It poses as one of the most important and growing threats to public health. Being an intravenous drug user (IVDU) is one of the most important risk factors in acquiring this infection.

Only a minority of patients are able to clear the virus, and so do not run the risk of developing HCV induced end liver damage. The majority of patients will develop either acute or chronic hepatitis that will require treatment. PegIFN-α2a in combination with ribavirin was used as a standard treatment in patient that did not meet exclusion criteria. Since 2011, a few generations of direct-acting antivirals (DAAs) were approved for HCV treatment (*Zeuzem et al., 2011*). Each of three DAAs subgroups can be used as a component of a triple combination regimen with PegIFN-a and ribavirin, yielding sustained viral response (SVR) rates of 60–100%. Response rate depends on the DAA used, the HCV genotype, the presence of detectable pre-existing amino acid substitutions conferring resistance to the DAA used and the severity of liver disease (*European Association for the Study of the Liver, 2014*). With three new HCV DAAs approved, IFN-free combinations are reserved for patients with advanced liver disease (fibrosis METAVIR score F3 or F4).

Among the different and diverse list of factors that influence the therapeutic response, the host's cytokines play a very important role. The cytokine levels are directly influenced by certain gene polymorphisms located within their coding or regulatory regions (*Wilson et al., 1997*).

Among others, IL-6 is reported to be elevated in chronic HCV infection compared to healthy controls (*Malaguarnera et al., 1997*). The low-producing interleukin-6 genotype IL-6 CC (IL-6 rs1800795 G174C) was associated with spontaneous clearance of HCV in patients infected by contaminated blood products (*Barrett et al., 2001*). On contrary, high producing interleukin-6 genotypes of the rs1800795 174G/C polymorphism (i.e., GG or GC genotypes) were associated with a greater likelihood of SVR in patients coinfected with HCV and HIV (*Nattermann et al., 2007*).

The human interleukin-28B gene encodes interferon lambda-3 (IFN-lambda-3). Interferon lambda (IFN-λ) has demonstrated antiviral activity against HCV genotype 1 in vivo (*Pagliaccetti & Robek, 2010*) and in vitro (*Muir et al., 2010*). It has been shown that a single nucleotide polymorphism (SNP) of IL28B gene (IL-28B rs12979860 C/T) predicts hepatitis C treatment induced viral clearance (*Ge et al., 2009*; *Halfon et al., 2011*; *Lin et al., 2011*; *Lindh et al., 2011*; *Luo et al., 2013*) and is associated with spontaneous resolution of hepatitis C infection (*Duggal et al., 2013*; *Kurbanov et al., 2011*; *Shi et al., 2012*; *Thomas et al., 2009*; *Tillmann et al., 2010*).

Here, we studied whether interleukin-28B or interleukin-6 (IL-6) promoter SNP affects the response to the PegIFN-α2a/ribavirin antiviral treatment in IVDU patients diagnosed with chronic hepatitis C. We determined IL-6 promoter and IL-28B gene polymorphisms in a cohort of 110 patients with chronic hepatitis C. All IVDU positive patients diagnosed with chronic hepatitis C were treated with a standard protocol of peg-interferon alpha-2a and ribavirin. Rates of SVR were compared between IL-6 and IL-28B wild type, heterozygous and homozygous genotypes.

## SUBJECTS AND METHODS

### The study samples

The sample of 112 patients was recruited from the outpatient hepatology unit at the Split Medical Center, Croatia and was followed from September 2007 until November 2013. The sample was drawn from the total of 947 HCV positive patients but only 112 individuals were diagnosed, biopsied, genotyped, treated and followed for a period of four years. Two patients were excluded from further analysis due to an ambiguous SVR status, possible reuse of intravenous drugs and re-infection with a different viral strain, resulting in a cohort of 110 patients. All enrolled patients had an alcohol consumption of less than 14 units per week and other common forms of chronic liver disease were excluded in all cases. Patients were Caucasians with the median (interquartile ranges (IQR)) age at diagnosis of 40 (35–45) years in the SVR group and 41.5 (39–47) in the non-responder (NR) group.

We also used the data from the 10,001 Dalmatians biobank as the source of population-based sample of the underlying, general Croatian population. All controls were apparently healthy subjects with no record of addiction, risky behaviors or detected HCV infection (*Polašek, 2013*; *Rudan et al., 2009*).

Patients' data were collected as a part of standard clinical procedure and the informed consent was obtained prior to participating in the study in all cases. The Ethics Committee of the Clinical Hospital Split approved the study (No: 2181-147-06-01/01-M.J).

### Diagnosis and treatment of HCV infection

A third generation enzyme immunoassay (ELISA; Abbott Diagnostics, Wetzlar, Germany) was used to test all subjects for HCV specific antibodies. Reverse-transcriptase polymerase chain reaction (RT-PCR) assay (Amplicor; Roche Diagnostic Systems, New Jersey, USA) was used to test for HCV RNA in all subjects, in order to determine SVR. PegIFN-α was used as a subcutaneous injection of 180 μg (or less if dose reduction was needed), once a week. Depending on patient's body weight ($\leq 75 \geq$ kg), a total of 1,000 or 1,200 mg of ribavirin was administered, in divided doses.

SVR was defined as undetectable HCV RNA 12 weeks or 24 weeks after treatment completion as assessed by a sensitive molecular method with a lower level of detection 15 IU/ml. NRs were patient who did not meet the SVR.

Our patients were followed up for four years and we were able to identify relapses. Patients that relapsed during the follow up time were included in NR group.

## Liver histology

Percutaneous liver biopsy was performed at the time of initial diagnosis and at the beginning of the treatment, using the Trucut biopsy technique (Sterylab, Rho MI, Italy) following informed consent. Inflammation was graded using a histological activity index (HAI) (*Knodell et al., 1981*) and fibrosis (*Ishak et al., 1995*). Every fifth biopsy was independently validated by two pathologists. A minority of patients was not biopsied either due to secondary coagulopathy or refusal to sign the informed consent and to participate in the procedure.

## DNA extraction

A salting out technique was used to extract DNA from whole blood or using the QIAmp DNA midi prep kit, (Qiagen Ltd., Crawley, UK). The obtained DNA was used for IL-6 promoter and IL-28B genotyping. During this process, all the RNA was removed by incubating the digested preparation with 1.5 ml ribonuclease A (Boehringer Mannheim UK Ltd, East Sussex, UK) per 400 ml of nuclear lysate.

## IL-6 and IL28B genotyping

A 175 base pair (bp) fragment of the human IL-6 gene spanning the promoter IL-6 G-174C region was amplified with gene specific primers (Roche Diagnostics, Alameda, California, USA). The resulting PCR fragments were analyzed with hybridization probes labeled with LightCycler Red 640. The genotypes were identified by running a melting curve with specific melting points (Tm) (Tib Molbiol GmbH, Berlin, Germany).

Genetic polymorphism in a SNP located near the IL-28B gene (rs12979860) was determined by enzymatic digestion of the PCR product. DNA fragments were amplified by using specific primers. Primer sequences were 5′-GCCTGTCGTGTACTGAACCA-3′, and 5′-GCTCAGGGGTCAAATCACAGAAG-3′, a PCR product of 143 bp was digested with HhaI enzyme and the resulting fragments of 27, 38, 65 and 78 bp were separated on 20% acrylamide gel followed by silver staining (Fig. 1).

## Statistical analysis

Data were analysed with the statistical package SPSS 19.0 (IBM Corp, Armonk, NY, USA) and PLINK v1.07 software (*Purcell et al., 2007*). Absolute numbers and percentages were used to describe categorical data, whereas median and IQR were used to describe quantitative data. One sample binomial test was used to assess distributions of sex, and response to the treatment. The association of response-to-treatment with sex or the viral genotype in infected patients was estimated with the Pearson's chi square test. Furthermore, nonparametric Mann-Whitney test was used to assess age differences, or differences in severity of fibrosis between the response groups.

Genetic association tests were mostly performed within PLINK software by using the case and control design. We defined cases as the group of patients that achieved SVR. Full model association tests were run in PLINK for each SNP using either chi-square or, when appropriate, Fisher exact test; and the best-fit model was identified. Full model included basic allelic, Cochran-Armitage trend, genotypic, dominant gene action and the

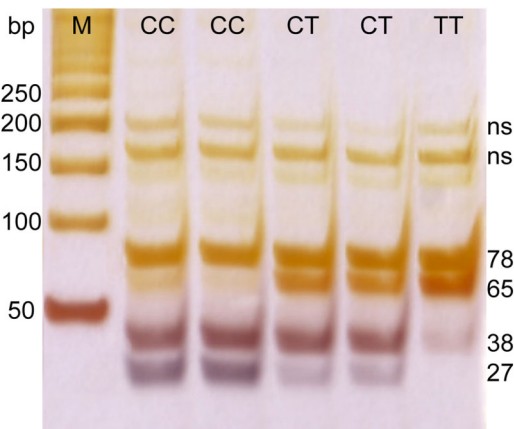

**Figure 1 PCR electrophoresis results for rs12979860 genotyping.** M, marker; ns, non specific band; CC, CT, and TT–rs12979860 genotypes.

recessive gene action tests. Additionally, the difference in distribution of IL28B CC carriers vs. CT+TT carriers across the response groups was estimated by the Pearson's chi square test using the SPSS v19 software (IBM, Armonk, NY, USA) since the recessive model in PLINK tests only the minor allele and in both our samples (patients and the underlying, healthy general population) the allele C at the IL28B locus was the dominant allele. All significant p-values yielded by genetic tests were further controlled by empirical p-values which were generated by the permutation procedure. Cochran-Mantel-Haenszel statistics was used to test whether the predictive power of the SNP markers was independent of viral genotypes detected in infected patients. Associations of the SVR status with both SNPs were further evaluated by multivariate logistic regression while accounting for covariates/factors: age, viral genotype, stage of fibrosis.

The achieved post hoc power for genetic association tests was calculated with Genetic Power Calculation software (http://pngu.mgh.harvard.edu/~purcell/gpc/), whereas G*power version 3.1.7 (Universität Kiel, Kiel, Germany) was used to assess the achieved power for comparison of genotype frequencies between the two population samples.

## RESULTS

A total of 110 IVDU patients with elevated liver function tests were diagnosed with a chronic hepatitis C, treated by standard interferon 2-alfa/ribavirin protocol and followed over at least one year for a SVR. The majority of treated patients (78 out of 110, or 71%) achieved SVR (one sample binomial test, $p < 0.001$). Men were significantly more prevalent in our sample than women: 82 (75%) vs. 28 (25%) (one sample binomial test, $p < 0.001$). Demographic and clinical characteristics of the sample are presented in Table 1. As shown in Table 1, NRs were significantly older than responders. We found no association of the SVR status with the sex, the initial Ishak score fibrosis stage, or the viral genotype.

Distributions of patients' genotypes across response-to-treatment groups are shown for both SNP loci, rs1800795-IL6 and rs12979860-IL28B, in Table 2.

**Table 1 Demographic and clinical characteristics of IVDU patients by the response-to-treatment status.**

|  | SVR, N = 78 | NR, N = 32 | Statistics |
|---|---|---|---|
| **Sex, N (%)** | | | |
| Men | 56 (72%) | 26 (81%) | Chi square test, p = 0.301 |
| Women | 22 (28%) | 6 (19%) | |
| **Age, median (IQR)** | 40 (35–45) | 41.5 (39–47) | Mann-Whitney, p = 0.041 |
| **Fibrosis stage, N (%)** | | | |
| Mean Ishak score[†] | 3 (2.0–3.0) | 2 (1.0–3.0) | Mann-Whitney, p = 0.077 |
| Mean Knodell score[‡] | 7 (5.0–9.0) | 6 (5.0–9.0) | Mann-Whitney, p = 0.803 |
| **Viral genotype, N (%)** | | | |
| 1 | 51 (73%) | 19 (27%) | Chi square test, p = 0.706 |
| 3 | 27 (68%) | 13 (33%) | |

Notes:
SVR, sustained viral response; NR, non responders.
[†] Liver fibrosis was scored by Ishak fibrosis score (0–6) as previously described (*Ishak et al., 1995*).
[‡] Histology activity index by Knodell score (0–18) in chronic active hepatitis (*Knodell et al., 1981*).

**Table 2 Distributions of rs1800795-IL6 and rs12979860-IL28B genotypes, by the response-to-treatment group.**

| SNP | Genotype | SVR, N = 78 | NR, N = 32 | Total |
|---|---|---|---|---|
| IL-28B, N (%) | CC | 22 (28%) | 8 (25%) | 30 (27%) |
| | CT | 50 (64%) | 19 (59%) | 69 (63%) |
| | TT | 6 (8%) | 5 (16%) | 11 (10%) |
| IL-6, N (%) | GG | 27 (35%) | 16 (50%) | 43 (39%) |
| | GC | 37 (47%) | 15 (47%) | 52 (47%) |
| | CC | 14 (18%) | 1 (3%) | 15 (14%) |

Note:
SVR, sustained viral response; NR, non responders.

With regard to genetic variation at the rs1800795-IL6 locus, we determined Hardy-Weinberg equilibrium in both response groups (exact test, p-values from 0.392 to 0.819). Overall, the genetic association tests indicated that the addition of allele 'C' has protective effect and increases the chance of achieving SVR. Specifically, several genetic association tests confirmed the association between rs1800795-IL6 polymorphism and SVR status, with the Cochran-Armitage trend test providing the best model fit ($\chi$ = 4.477, df = 1, p = 0.034, empirical p = 0.039). Patients with rs1800795-IL6 CC genotype had significantly better SVR (14 out of 15, 93%) compared to those with GC (37 out of 52, 71%) or GG (27 out of 43, 63%) genotypes (Fig. 2). The achieved post-hoc power of this association test was high: 90% (calculated under the additive model, at type I error rate of 0.05 and with responders/NRs ratio of 78/32).

After controlling for viral genotypes, the association of rs1800795-IL6 polymorphism with SVR status remained significant (Mantel-Haenszel 2 × 2 $\chi^2$ = 4.483, p = 0.034, with odds ratio of 1.99, 95% CI 1.05–3.78). The association was further evaluated by multivariate logistic regression analysis while accounting for covariates: age, viral genotype, and fibrosis (F ≥ 3). The result of regression analysis confirmed that the

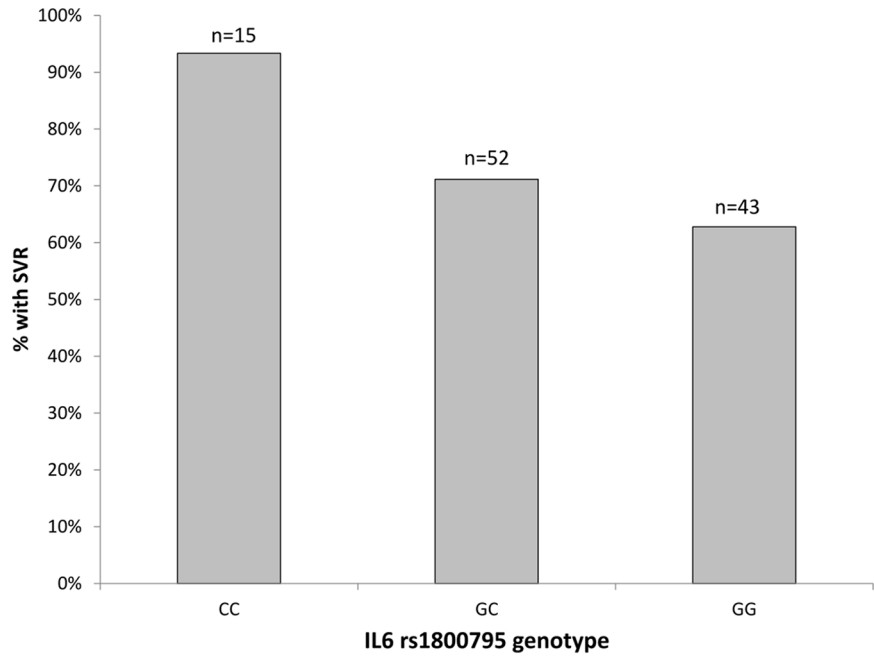

**Figure 2 The percentage of patients with SVR by the rs1800795-IL6 genotypes.** n, total number of patients with each genotype.

addition of allele 'C' increased the chance of achieving SVR (OR 2.45, 95% CI 1.13–5.30, p = 0.023). Besides the rs1800795-IL6 polymorphism, age was the only covariate that significantly affected SVR status, with each additional year slightly decreasing the chance of SVR response (significance at the level of 0.1, OR 0.95, 95% CI 0.89–1.01, p = 0.096).

The genotype frequencies at the rs12979860-IL28B locus met the Hardy-Weinberg expectations in nonresponders (exact test, p = 0.473), and deviated from the Hardy-Weinberg equilibrium in SVR cases (i.e. SVR responders, p = 0.004). When we analysed the genotype frequencies at the rs12979860 locus in the IVDU cohort, and in the underlying, apparently healthy, general-population sample (n = 531 individuals; HWE, p = 0.999), we observed that the frequencies of CC, CT and TT genotypes in our cohort were: 27, 63, and 10%; whereas the corresponding frequencies in the population-based sample were 49, 42, and 9% (Fig. 3). The finding demonstrated a large and significant reduction of CC genotype (z-score test for proportions, p < 0.001), and a significant increase in frequency of heterozygous TC genotype (p < 0.001) in the IVDU cohort of chronic HCV patients compared to the underlying, apparently healthy population. The frequencies of TT genotype were comparable between the groups (p = 0.741). The post hoc power for detecting medium size effect in genotype frequencies between these two population samples was 99%.

With regard to the SVR responder and NR groups, we did not, however, observe a significant association of SVR response with the allele-carrier groups: CC and CT+TT ($\chi^2$ = 0.118, df = 1, p = 0.732); although the frequency of CC genotype was in fact somewhat higher in responders (by 3%). The result persisted after we controlled for viral

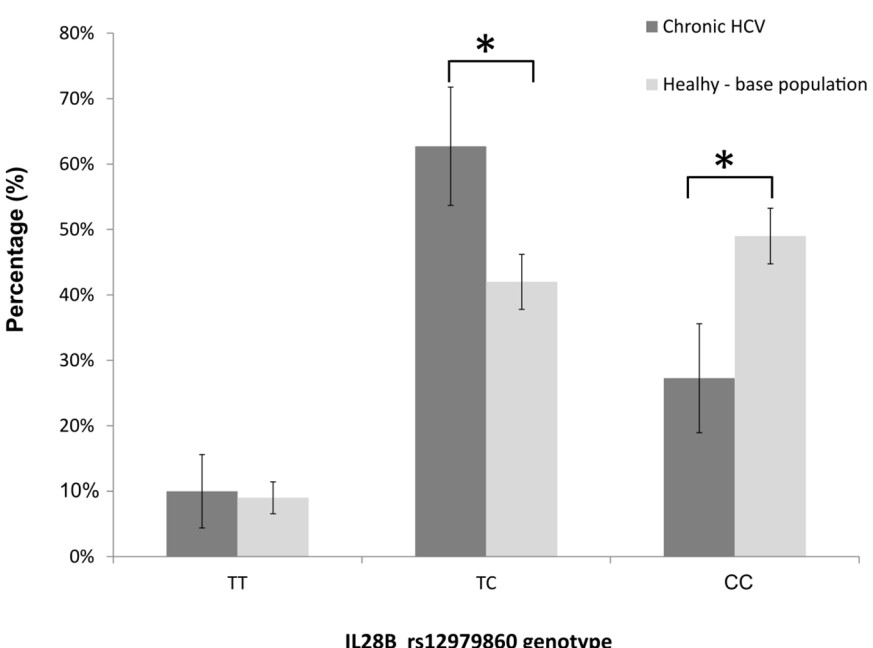

**Figure 3 The distribution of rs12979860-IL28B genotypes among: IVDU with chronic HCV (in black bars), and the underlying (base), apparently healthy population (in gray bars).** Shown are percentages with accompanying 95% CI. * Statistically significant difference, p < 0.05.

genotypes (Mantel-Haenszel $2 \times 2$ $\chi^2$ = 0.630, p = 0.427) and after multivariate logistic regression using age, viral genotype and fibrosis stage as additional predictors. It should be noted, however, that the achieved power for this test, under the dominant genetic model: CT+TT vs. CC, type I error rate of 0.05, and observed responders/NRs ratio was very low: 9%. This was in contrast to a priori power calculation that was based on Ge's data (*Ge et al., 2009*) which showed that in treatment-naive, chronic HCV type 1 patients of European-American origin, as little as 20 patients were enough to achieve 80% of power for the rs12979860-IL28B association test on the response to therapy. Also, the observed penetrance of NR phenotype in our cohort of 29% was considerably lower than the penetrance in Ge's cohort of 45% (z-score test for proportions, p = 0.001; achieved power 80%).

## DISCUSSION

This study investigated the role of IL-6 and IL-28B gene polymorphisms on SVR in IVDU patients diagnosed with chronic hepatitis C infection. The treatment was conventional and included Peg IFN combined with ribavirin for either 48 weeks (genotype 1) or 24 weeks (genotype 3). We defined sustained virological response (SVR) as an absence of detectable virus at the end of follow up evaluation and or disease relapses according to the standard definitions (*Ghany et al., 2009*).

Interleukin-6 was originally discovered as a protein that caused the final differentiation of B cells into immunoglobulin secreting cells (*Muraguchi et al., 1988*). Additional work showed that IL-6 and its receptor—sIL-6Rα suppress neutrophil recruitment at site of acute inflammation, making way for the influx of monocytes as the inflammatory

response is sustained (*Kopf et al., 1994*). IL-6 is well known pro-inflammatory cytokine with pro-tumorigenic potential (*Grivennikov et al., 2009*) and is emerging as a key regulatory signal in the development of the newly described pro-inflammatory effectors T-cell subset, so called Th17 cells (*Harrington et al., 2005*). The IL-6 rs1800795 G allele has been also associated with higher degrees of liver necroinflammation (*Falleti et al., 2010*) and fibrosis (*Cussigh et al., 2011*).

In our study, allele C at rs1800795-IL6—a SNP in the IL-6 gene promoter, was associated with SVR (OR 2.45, 95% CI 1.13–5.30, p = 0.023). The genotype that confers the highest degree of protection in terms of achieving SVR, rs1800795-IL6 CC, demonstrated an overwhelming lower relapse rate in HCV treated patients (1 out of 15 patients, 7%). Similar results were reported for Italian non-IVDU HCV infected patients thus corroborating the importance of our findings (*Cussigh et al., 2011*). According to prior studies, the CC genotype appears to be associated with significantly lower level of plasma IL-6, whereas the GG and GC genotypes appear to have higher levels of plasma IL-6 (*Fishman et al., 1998*; *Lapiński, 2001*). This implies a possible connection of IL-6 status with the therapy outcome. The putative low producing IL-6 phenotype may play a protective role against chronic hepatitis C infection by helping to clear the viral particles during standard therapy. Chronic hepatitis C patients with rs1800795-IL6 CC genotype and lower IL-6 serum level may have an attenuated adoptive immune response, directed away from damaging, pro-inflammatory and autoimmune to predominately suppressive and anti-viral inflammatory response.

An IL-28B gene SNP is located 8 kb upstream of the start codon of IL-28B gene that encodes IFN-$\lambda$ a member of type III IFN family. IFN-$\lambda$ interacts with a transmembrane receptor to induce a potent antiviral response (*Donnelly & Kotenko, 2010*; *Fox, Sheppard & O'Hara, 2009*; *Li & Huang, 2007*). The antiviral activity is mediated through the activation of the either JAK-STAT (IFN $\alpha$, $\lambda$ and $\gamma$) or MAPK (IFN $\alpha$ and $\lambda$) pathways (*Arslani et al., 2013*). There is a strong association of genetic variations in IL-28B gene with response to therapy (*Ge et al., 2009*; *Suppiah et al., 2009*; *Tanaka et al., 2009*), and with spontaneous HCV clearance (*Duggal et al., 2013*). In our cohort of IVDU patients diagnosed with chronic hepatitis C, no association between the therapy outcome and the SNP in the IL-28B gene, rs12979860-IL28B was identified, but the study was largely underpowered to draw a solid conclusion from this test. Nevertheless, our data still do support the involvement of the rs12979860-IL28B CC genotype in both of these patho-physiological/immunological processes: spontaneous HCV clearance and the response to therapy.

Firstly, when we compared our cohort of IVDU patients suffering from chronic HCV with the population-based sample taken from the apparently healthy, underlying population; the frequency of the favorable rs12979860-IL28B CC genotype which has been also associated with the spontaneous clearance of HCV, was largely decreased in the patient group. Conversely, the frequencies of genotypes considered to have a neutral effect on acquiring HCV infection/response to therapy were either increased in patients (TC) or showed no difference between the samples (TT). In other words, the protective genotype rs12979860-IL28B CC was found at much lower frequency in infected IVDU individuals with chronic HCV (27%) than in the underlying, healthy population

(49%), thus strongly pointing towards the selective loss of CC homozygotes in the patient population.

In line with our finding, *Gélinas et al. (2013)* have observed significantly higher prevalence of the responder genotype rs12979860 CC in a group of IVDU who were spontaneous resolvers from a HCV infection than in a baseline population of IVDU users; suggesting a dilution of CC genotype in chronic IVDU HCV patients. Additionally, also supporting our findings, *Ezzikouri et al. (2013)* found that patients who had cleared HCV spontaneously were from 2.7 to 4.7 times more likely to carry CC genotype than the TC, or the TT genotype, respectively, while *Montes-Cano et al. (2010)* observed the CC genotype in 73% of individuals with spontaneous resolution of HCV infection versus only 46% in individuals with the persistent infection.

Secondly, our results still suggest the positive effect of rs12979860-IL28B CC genotype in acquiring SVR. In particular, the distribution of rs12979860-IL28B genotypes in the cohort of IVDU chronic patients significantly deviated from HWE-P only in the group of responders, whereas in NRs, and in healthy controls the rs12979860-IL28B genotypes followed the HWE. Since the case (SVR responder) genotypes will only be in HWE under the multiplicative genetic model (*Lewis & Knight, 2012*), the departure from this equilibrium, if found exclusively in cases, can be expected in relatively small samples of patients over a range of genetic models and is indicative of the actual association to the trait under study (*Wittke-Thompson, Pluzhnikov & Cox, 2005*). In addition, we have observed somewhat higher percentage of CC genotype in SVR cases, although this percentage did not reach a statistical significance.

Similar to our results, *Seaberg et al. (2015)* also did not find an association between rs12979860-IL28B genotypes and the spontaneous clearance of HCV in men who have sex with men. There are obviously a large number of factors: demographic, viral, and human genetic factors; which influence HCV viremia and the results on distribution of particular SNPs should be interpreted in a larger context. The fact that the frequency of rs12979860 CC genotype varies in different ethnic groups or geographical areas adds to this complexity. In particular, several studies have estimated that East Asians had a high percentage of CC genotype, whereas the frequency of this genotype was intermediate in Europeans and minor frequency in African cohorts (*Ge et al., 2009*; *Thomas et al., 2009*).

Concerning the geographic variability of rs12979860 CC genotypes, previous studies on Caucasian patients who were infected with the HCV viral genotype 1 estimated the prevalence of CC genotypes to be between 35 and 39% (*Cavalcante et al., 2012*; *Ge et al., 2009*; *Montes-Cano et al., 2010*; *Nattermann et al., 2011*). The percentage of CC genotypes observed in our patients infected with the same viral genotype was 27% (95% CI 19–36%) which is somewhat lower, suggesting that the impact of IL-28B polymorphism on acquiring of infection or spontaneous clearance of HCV might be more prominent in the Croatian population. In addition, it seems that compared to *Ge et al.'s (2009)* cohort of chronically infected HCV patients, penetrance of nonresponders in our IVDU cohort exhibited considerably lower value. This might indicate that the genetic background of the Croatian population is such that both the HCV spontaneous clearance and the response to

therapy in chronically infected IVDU is more pronounced in this population. Having in mind that the population differences in rates of spontaneous clearance have also been proposed for IVDU patients (*Fischer et al., 2004*; *Miedzinski & Taylor, 2008*), the impact of IL-28B polymorphism on the spontaneous clearance of HCV in the Croatian population should be investigated in more details in order to increase our knowledge on the therapeutic effectiveness of PegIFN-α2a/ribavirin on rs12979860 genotypes in different populations, particularly in high-HCV-risk IVDU population.

## CONCLUSIONS

In conclusion, we have identified that the rs1800795-IL6 CC genotype is associated with significantly better SVR to the standard Peg IFN and ribavirin treatment in IVDU/HCV patients. Also, findings point towards a strong protective role of rs12979860-IL28B CC genotype in acquiring chronic hepatitis C infection in the Croatian IVDU population. Finally, among all covariates, age is the most important, where every additional year slightly decreases the chance of SVR response. Further prospective and large scale clinical studies are necessary to confirm our results before we can prospectively identify IVDU and HCV patients for whom therapy is likely to be successful.

## ACKNOWLEDGEMENTS

We thank our patients for taking part in this study, and Mrs. Sandra Vujević for technical assistance.

### Funding

The project was funded by the Croatian Ministry of Science, grant number 258-2160800-0333 to Š.A. J.T. was supported by the Croatian Science Foundation grant No: IP-2014-09-1904 and the Croatian Ministry of Science, Sport and Education grant number 216-000000-3348. The funders had no role in study design, data collection and analysis, decision to publish, or preparation of the manuscript.

### Grant Disclosures

The following grant information was disclosed by the authors:
Croatian Ministry of Science: 258-2160800-0333.
Croatian Science Foundation: IP-2014-09-1904.
Croatian Ministry of Science, Sport and Education: 216-000000-3348.

### Competing Interests

The authors declare that they have no competing interests.

### Author Contributions

- Zoran Bogdanović conceived and designed the experiments, performed the experiments, analyzed the data, contributed reagents/materials/analysis tools, wrote the paper, prepared figures and/or tables, reviewed drafts of the paper.

- Ivana Marinović-Terzić performed the experiments.
- Sendi Kuret performed the experiments, contributed reagents/materials/analysis tools.
- Ana Jerončić analyzed the data, wrote the paper, prepared figures and/or tables, reviewed drafts of the paper.
- Nikola Bradarić contributed reagents/materials/analysis tools.
- Gea Forempoher contributed reagents/materials/analysis tools.
- Ozren Polašek contributed reagents/materials/analysis tools.
- Šimun Anđelinović contributed reagents/materials/analysis tools.
- Janoš Terzić conceived and designed the experiments, analyzed the data, contributed reagents/materials/analysis tools, wrote the paper, reviewed drafts of the paper.

## Human Ethics

The following information was supplied relating to ethical approvals (i.e., approving body and any reference numbers):

The Ethics Committee of the Clinical Hospital Split No: 2181-147-06-01/01-M.J.

## Data Deposition

The raw data has been supplied as Supplemental Dataset Files.

## Supplemental Information

Supplemental information for this article can be found online at http://dx.doi.org/10.7717/peerj.2576#supplemental-information.

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
