# Peer review of "The impact of IL-6 and IL-28B gene polymorphisms on treatment outcome of chronic hepatitis C infection among intravenous drug users in Croatia"

_PeerJ, doi:10.7717/peerj.2576_

## Round 0.1 · original submission · Major Revisions

In my opinion the crucial improvement that needs to be done in this study is the increase of the number of patients when presenting gene polymorphism-related conclusions. This was also pointed out by the reviewer. I hope you will find all the recommendations helpful.

·

Basic reporting

Zoran Bogdanović and colleagues investigated the possible influence of the suggested IL-6 and Il-28 genetic biomarkers on the progression of HCV disease in crotaian patients in the study entitled “The impact of IL-6 and IL-28B gene polymorphisms on treatment outcome of chronic hepatitis C infection
among intravenous drug users in Croatia”. The authors discovered that Patients showed a significantly better response to treatment according to the number of copies of the C allele carried at rs1800795-IL6.


I appreciate the time and effort put to this study by the authors; however there is one major concern about this study which is that the number of patients used in this study is a bit small to draw solid conclusions out of the study. Therefore, I recommend increasing the number of patients to at least 150-200 patients.
-Minor issues: include an agarose gel picture for the IL-28 gene.

Experimental design

the study design is ok.

Validity of the findings

No comments

Additional comments

No comments

·

Basic reporting

Introduce treatment options in "Introduction". Patients in this work used standard therapy based in pegylated interferon and ribavirin. However, new direct acting antivirals (DAAs) were introduced to hepatitis C therapy and must to be cited.
More references must to be included in some specific parts (there are notes on "track changes" .doc file).

Experimental design

Describe response type classification on "Methods". Comment if relapser patients were included in non-responder group.

Validity of the findings

No comments

Additional comments

This article is relevant even though another groups reported these SNPs association. The article is well writen but must to be revised. Most important points are highlighted in the attached file.

---

## Round 0.2 · accepted · Accept

Dear Ana,
Thank you for your submission to PeerJ. Your manuscript has been accepted for publication. Congratulations!

Krystyna Dąbrowska